# Characterization of Static Strength, Vertical Jumping, and Isokinetic Strength in Soccer Players According to Age, Competitive Level, and Field Position

**DOI:** 10.3390/ijerph20031799

**Published:** 2023-01-18

**Authors:** João Nuno Gouveia, Cíntia França, Francisco Martins, Ricardo Henriques, Marcelo de Maio Nascimento, Andreas Ihle, Hugo Sarmento, Krzysztof Przednowek, Diogo Martinho, Élvio Rúbio Gouveia

**Affiliations:** 1Department of Physical Education and Sport, University of Madeira, 9020-105 Funchal, Portugal; 2Marítimo da Madeira—Futebol, SAD, 9020-208 Funchal, Portugal; 3LARSYS, Interactive Technologies Institute, 9020-105 Funchal, Portugal; 4Research Center in Sports Sciences, Health Sciences, and Human Development (CIDESD), 5000-801 Vila Real, Portugal; 5Department of Physical Education, Federal University of Vale do São Francisco, Petrolina 56304-917, Brazil; 6Department of Psychology, University of Geneva, 1205 Geneva, Switzerland; 7Center for the Interdisciplinary Study of Gerontology and Vulnerability, University of Geneva, 1205 Geneva, Switzerland; 8Swiss National Centre of Competence in Research LIVES—Overcoming Vulnerability: Life Course Perspectives, 1015 Lausanne, Switzerland; 9University of Coimbra, Research Unit for Sport and Physical Education (CIDAF), Faculty of Sport Sciences and Physical Education, 3004-504Coimbra, Portugal; 10Institute of Physical Culture Sciences, Medical College, University of Rzeszów, 35-959 Rzeszów, Poland

**Keywords:** muscle strength, peak torque, total work, average power, countermovement jump, squat jump, football

## Abstract

Muscular strength is strongly related to speed and agility tasks, which have been described as the most decisive actions preceding goals in the soccer game. This study aimed to characterize the players’ strength indicators and to analyze the variation associated with age, competition level, and positional role. Eighty-three male soccer players from A team (*n* = 22), B team (*n* = 17), U-23 (*n* = 19), and U-19 (*n* = 25) participated in this study. Handgrip strength was assessed using a hand dynamometer (Jamar Plus+), countermovement jump (CMJ) and the squat jump (SJ) were evaluated using Optojump Next, and a Biodex System 4 Pro Dynamometer was used to assess the isokinetic strength of knee extension/flexion. Team A players showed increased lower-body strength compared to their peers, mainly through their increased vertical jumping capacity (i.e., CMJ and SJ; ps < 0.019), and superior performance in isokinetic assessments (i.e., peak torque, total work, and average power). Overall, older players outperformed their younger peers regarding vertical jumping, static strength, and average power in isokinetic strength (ps < 0.005). This study emphasizes the superior strength levels of professional soccer players compared with their lower-division peers, even after controlling by age. This information is of great value to sports agents and coaches, underlining the need to design and include strength-specific content during soccer training.

## 1. Introduction

In soccer, high-speed and intense actions, such as jumping, sprinting, and direction changes, have been described as the most decisive actions preceding goals [1]. Despite the dominance of aerobic metabolism during match performance, muscle strength has a significant role in the force–time characteristics related to soccer performance, particularly its strong relationship with high-speed and high-intensity actions [2,3]. Besides the influence of muscular strength on various factors associated with athletic performance, greater muscular strength has been described as an important factor in decreasing the risk of injury [4]. For this reason, sports agents and coaches worldwide have assessed players’ physical attributes to identify their risk of injury and plan specific training content for the season according to their profiles.

In the literature, “muscle strength” refers to maximal muscular force in a single voluntary contraction [5]. On the other hand, “muscle power”, commonly defined as “torque”, is related to the ability of a force to cause rotation on a lever [6]. The isokinetic dynamometer has emerged as the gold standard for evaluating static and dynamic muscle function in rehabilitation and sports environments [7,8]. This equipment accommodates resistance and implies a constant angular velocity [6]. Dynamic strength may be the most common measure of an individual’s strength among testing options. The concentric muscular action (CC) occurs when the tension developed by the muscle is superior to the resistance it must overcome, and a shortening happens. These actions commonly emerge in the positive phase of most strength training exercises and may contribute to the performance of speed and change of direction tasks [9].

However, the literature has also described the isokinetic assessment as a laboratory and analytical testing that does not reflect the functional aspects of limb movements involved in soccer performance [4]. Consequently, vertical jumping tasks, such as the squat jump (SJ) and the countermovement jump (CMJ), have been recommended for lower-body strength and muscular imbalance assessment [2,10,11]. Although demanding specific equipment for evaluation (such as force platforms and jump apps, among others), both the SJ and CMJ are considered easy and reliable field tests. The SJ evaluates the ability to rapidly develop force exclusively during the concentric movement, whereas the CMJ assesses the capability to produce force in the stretch–shortening cycle [12].

In soccer, a previous study comparing knee flexor muscle strength considering the players’ competitive level reported that elite players presented stronger hamstrings than their non-elite and amateur peers [13]. Additionally, the literature has described the superior jumping ability of elite (49.9 ± 7.5 cm) compared to non-elite soccer players (43.9 ± 6.9 cm) [14]. These results underline that individuals involved in the highest competitive levels tend to outperform their lower-divisions counterparts in lower-body strength assessments.

Meanwhile, handgrip strength has been widely used to predict absolute strength, which may be evaluated simply and cost-effectively. In college-aged individuals, a strong and positive correlation was observed between handgrip strength and the one-repetition maximum leg extension performance [15]. Across several sports contexts, when comparing players from different competitive levels, elite players have shown significantly better scores on handgrip strength performance than their peers [16]. Nevertheless, data on this topic among soccer players are lacking.

Overall, the soccer literature has widely explored research focused on strength assessment. However, data are limited when considering individuals’ characteristics such as age, competitive level, and field position. Indeed, some existing reports confirm a varied profile of isokinetic strength and vertical jumping performance between players in different field positions [17,18,19]. In contrast, although the handgrip strength is routinely used as a field test, few data are available on the relationship between its performance and the players’ profile. Therefore, this study aimed to characterize soccer players’ strength indicators and to analyze the variation in performance associated with age, competition level, and field position.

## 2. Materials and Methods

### 2.1. Participants

Eighty-three male soccer players from the first team (A team), second team (B team), under-23 team (U23), and under-19 team (U19) participated in this study. Specifically, 22 players were from the A team (age = 25.9 ± 4.2 years, height = 181.9 ± 6.3 cm, body mass = 78.1 ± 6.9 kg), 17 players from the B team (age = 22.3 ± 1.8 years, height = 180.9 ± 9.5 cm, body mass = 78.5 ± 11.6 kg), 19 players from the U23 group (age = 20.6 ± 1.5 years, height = 177.9 ± 6.0 cm, body mass = 72.9 ± 9.2 kg), and 25 players from the U19 group (age = 17.8 ± 0.7 years, height = 176.8 ± 6.4 cm, body mass = 70.6 ± 6.0 kg). Age was dichotomized into two categories according to the median (20.9). The A team was competing at the First Portuguese League, the B team was competing at the Fourth Portuguese National Division, and the U23 and U19 teams were competing in the main National Division for their age groups. Overall, players were from 14 different countries; however, most of them (*n* = 56) were Portuguese. All teams were part of the same club. The assessments were conducted at the beginning of the sports season, and all participants were healthy at the time of data collection. Participants were tested for static strength, vertical jumping, and muscular strength. During four consecutive days, each team was evaluated in a physical performance laboratory following the mentioned testing order. Between tests, a 5 min break was given to each participant to avoid the effects of accumulated fatigue.

All procedures applied were approved by the Ethics Committee of the Faculty of Human Kinetics, CEIFMH No. 34/2021. The investigation was conducted following the Declaration of Helsinki, and informed consent was obtained from the underage participants’ legal guardians. All test assessments were conducted by trained staff from the research team, who were experienced in each protocol.

### 2.2. Static Strength

The handgrip strength protocol consisted of three alternated data collection trials for each arm, performed using a hand dynamometer (Jamar Plus+, Chicago, IL, USA). The rest interval between each trial was set at 30 s. Participants held the dynamometer in one hand with the elbow at a 90° position and laterally to their trunk [20]. From this position, participants were asked to squeeze as hard as possible for about two seconds. If the participant’s arm was extended or if the dynamometer touched the body during execution, the trial was repeated. The best score of the three trials was used for analysis.

### 2.3. Vertical Jumping Assessment

The countermovement jump (CMJ) and the squat jump (SJ) were used to evaluate the vertical jumping capacity [21]. Participants performed four data collection trials 30 s apart. Although there are indications that endorse a 1 min passive rest between jumps to ensure muscular recovery [22], the issue is not consensual in the literature. In our study, due to time restrictions to perform all the assessments in the protocol, especially in the professional team, we considered a rest period of 30 s between the performance of the jump. Some studies have also presented shorter recovery times such as 20 s [23] and 30 s between each repetition of the CMJ [24]. The data were collected using the Optojump Next (Microgate, Bolzano, Italy) system of analysis and measurement. Participants were encouraged to jump to maximum height during testing. After the protocol explanation, three experimental trials were allowed for each participant to ensure correct execution.

For the CMJ, participants began standing, with feet placed hip-width to shoulder-width apart. Participants dropped into nearly 90° of knee flexion from this position, followed by a maximal-effort vertical jump. To avoid the influence of arm swing, the hands remained on the hips for the entire movement. The trial was repeated if excessive knee flexion was observed or if the hands were removed from the hips at any point. The participants reset to the starting position after each jump [25]. For the SJ, the participants started in a squat position of approximately 90° of knee flexion, followed by a maximal-effort vertical jump. The trial was repeated if a dipping movement of the hips was noticed. The participants reset to the starting position after each jump [25].

### 2.4. Isokinetic Assessment

The Biodex System 4 Pro Dynamometer (Shirley, NY, USA) was used to assess the isokinetic strength of knee extension and knee flexion from both lower limbs at two angular velocities: 60°/s and 300°/s. The chosen velocities vary between slow and fast. Velocities below 180°/s are considered slow [26]. Before data collection, participants performed a warm up on a reclining bicycle (Technogym Xt Pro 600 Recline, Cesena, Italy) for about 5 min. The effort varied between levels 4 and 5, while the cadence ranged between 50 and 60 rotations per minute. After the warm up, participants were seated on the dynamometer following the manufacturer’s guidelines, adopting a standardized position of 85° of hip flexion from the anatomical position. The lever arm of the dynamometer was aligned with the lateral epicondyle of the knee. The trunk, thigh, and leg (slightly above the medial malleolus) were stabilized with belts. The range of motion was defined as participants carrying the knee extension to its maximum range, corresponding to the full extension range in the isokinetic software. Then, participants were asked to flex the knee until 90° flexion. Since the force of gravity assists in the flexion phase and makes the extension phase more difficult, researchers performed an individual calibration for gravity correction before each assessment as recommended in previous literature [27]. This correction was carried out at the position of 30° of knee flexion. During the test, participants were instructed to keep their arms crossed with the hand on the opposite shoulder holding the belts [28]. During the assessment, enthusiastic verbal support was given to incentivize participants to “kick up” and “pull back” as hard and fast as possible. Five trials were performed at each speed before each isokinetic test to ensure the correct execution [29]. The sequence of the tests was defined as follows: (1) five repetitions of knee extension concentric–concentric combined with knee flexion concentric–concentric at 60°/s with 60 s intervals; (2) 10 repetitions of knee extension concentric–concentric combined with knee flexion concentric–concentric at 180°/s with 60 s interval; and (3) 10 repetitions of knee extension concentric–concentric combined with knee flexion concentric–concentric at 300°/s. This chosen sequence allows us to determine the maximum moment of strength, since for slower speeds, the targeted repetitions are between 4 and 5, and for faster speeds between 6 to 10 repetitions, with a rest time between sets of 60 to 160 s. The preferred leg was tested first, followed by the non-preferred leg under the same conditions and after a 3 min break.

The values of peak torque, total work, and average power were used for analysis. A more detailed description of these variables is presented in Table 1 according to the manufacturers’ manual [30].

### 2.5. Statistics

Descriptive statistics are presented as means (standard deviation). The Kruskal–Wallis test was used to investigate overall differences in static strength, vertical jumping, and isokinetic strength between groups and to verify differences in the performance according to the player’s field position. The Mann–Whitney test was conducted with Bonferroni adjustment to the alpha values to identify post hoc tests. The Mann–Whitney test was also used to investigate differences between older and younger players. Finally, a standard multiple regression was conducted to calculate the variance in each strength indicator that can be explained by age, competitive level, and player’s field position. All data analysis assumptions were verified. All analyses were performed using the IBM SPSS Statistics software 26.0 (SPSS Inc., Chicago, IL, USA). The significance level was set at *p* ≤ 0.05.

## 3. Results

Figure 1 depicts the static strength and vertical jumping performance according to players’ age. Regarding static strength, older players presented overall better results than their younger peers. The same trend was observed in the vertical jump assessment, only in the CMJ (*p* ≤ 0.001).

Figure 2 depicts the site-type-specific isokinetic strength concentric movement performances according to players’ age. After correcting by the Bonferroni method, with one expectation, 300°/s right knee extension, where older players performed better than the young ones (*p* < 0,001), non-significant differences were seen between age groups in the rest of the isokinetic strength concentric metrics. Older players showed significantly better results than younger players in the average power on left knee extension (60°/s and 300°/s), right knee extension (60°/s and 300°/s), and at 300°/s right knee flexion. No statistically significant differences were observed among older and younger players in the overall values of peak torque and total work. The bilateral differences in the peak torque in extension of the left knee vs. extension of the right knee and flexion of the left knee vs. flexion of the right knee were higher in younger (0.9–5.2%) compared with older players (0.2–3.8%).

Table 2 resumes the comparison between groups of the performance in static strength and vertical jumping considering the players’ competitive level. The A team presented significantly higher values than the U-23 and the U-19 groups regarding static strength. The same trend was observed in vertical jumping, with A team players considerably outperforming their peers.

Table 3 depicts the comparison between groups of the performance in isokinetic strength concentric metrics considering the players’ competitive level. In the overall PT and TW, the U-19 players showed significantly better results than the other groups, except for the right knee extension and knee flexion at 300°/s. In the average power, the A team players outperformed their peers, followed by the U-19 group.

Table 4 presents the results of static strength and vertical jumping according to players’ position. The goalkeepers and the defenders showed better performance levels in handgrip and jumping ability; however, the differences were not statistically significant.

Table 5 presents the results of the isokinetic strength performance according to field position. The attackers consistently presented slightly better scores than the midfielders in the variables assessed. Moreover, among all groups, the attackers showed higher mean scores regarding right knee flexion and extension at the different angular velocities evaluated. However, these overall differences were not statistically significant.

Finally, a standard multiple regression was conducted to calculate the variance in each strength indicator that can be explained by age, competitive level, and player’s field position. In the model, the competitive level was the strongest predictor of handgrip right side (beta = −0.63; *p* < 0.001), handgrip left side (beta = −0.60; *p* < 0.001), and CMJ height max (beta = −0.36; *p* = 0.029). The variance in each of these variables explained by the model ranged between 21% and 25%. Between isokinetic strength metrics, in the average power, again the competitive level led to the strongest variation on all metrics analyzed (−0.43 < betas > −0.76; ps < 0.004). The variance in each average power metric explained by the model ranged between 22% and 33%.

## 4. Discussion

In this study, players from the highest competitive level (A team) showed overall increased lower-body strength compared to their peers, mainly through their increased vertical jumping capacity and superior performance in isokinetic assessments, even after controlling for age. Overall, older players outperformed their younger peers regarding vertical jumping, static strength, and average power in isokinetic strength.

The literature reviewed has described the increased jumping ability of players in higher competitive levels compared to the ones competing in lower divisions [14,31,32]. This study corroborates this trend since the professional players’ group (A team) significantly outperformed their lower-division peers in the SJ as in the CMJ. Furthermore, the A team also showed substantially higher scores in handgrip, which is a strong indicator of overall strength [16]. These findings support the idea that explosive strength is one of the muscle skills that distinguishes elite teams from other teams that play in less competitive championships. Additionally, static or isometric strength should also be considered as a potential muscle skill that distinguishes the player’s level. These data open up new insights and reinforce the importance of accessing and including complementary strength training at all competitive levels to improve the muscular response to the competition.

Regarding isokinetic strength, the literature has pointed out that the value of PT at 60°/s during knee flexion and extension was higher in professional soccer players than in lower divisions [33,34]. In previous investigations conducted among professional soccer players, PT ranged between 237.3 N/m and 283.4 N/m on 60°/s knee extension and between 174.4 N/m and 177.8 N/m on 60°/s knee flexion [33,34], which is slightly higher than our results for the same type of sample.

Meanwhile, older players (age > 20.9 years) showed significantly higher scores in the vertical jumping tasks than younger players. Indeed, previous research has supported gains in leg muscle power with development in soccer players [35]. The analysis of the performance in isokinetic strength also supports the superior level of leg power among older players, namely in average power (in all angular velocities assessed), which is an indicator of explosive strength [11]. According to the literature, explosive strength is related to the ability to increase the force from a low or resting level [36]. In contrast, maximum strength is related to the greatest force a muscle group can produce [37]. Although maximum strength may be a foundation to develop explosive strength, the players’ training process could promote content focused on enhancing one’s individual capacity (e.g., using plyometric training to promote lower-body explosive strength).

Although previous literature has mentioned that the specific functional activity of players according to their field position influences the varied profile of isokinetic strength performance [38], our analyses did not show substantial differences between groups. A study conducted on 111 elite international players of the top division in Poland concluded that goalkeepers and central midfielders had lower PT strength levels at 60°/s compared to other players’ positions [38]. Different results were achieved by [39], reporting that defenders, midfielders, and attackers present similar jumping abilities, while goalkeepers outperform their peers [39]. Partially, our results are in line with this study. Although the differences were not statistically significant, a superior jumping ability of both goalkeepers and defenders was observed, while the attackers presented worse performance levels. We recognize that the literature on this topic is still lacking, which underlines the need for future works designed to assess the profile of isokinetic strength indicators according to field position. However, since muscular strength is strongly related to speed and agility tasks [2,3], which is a determinant for success in the game, it is highly recommended to consider specific field functions/positions’ lower-body explosive strength development during the soccer players’ training process.

The cross-sectional design and the lack of information regarding each group’s training process represent some of this study’s limitations. Moreover, only data concerning the concentric contraction mode were evaluated. The analysis of eccentric contraction values would provide a far more informative report of the muscle and the functional ratio that contemplates eccentric flexion PT and concentric extension PT [40]. Finally, we did not control the lateral dominance in the isokinetic assessments, which is critical because right and left values should not be interpreted in the same way. To deal with this limitation, we deeply analyzed the bilateral differences considering age groups (Table 3). Considering the peak torque measurement, we conclude that the bilateral differences ranged between 0.2% to 5.2%, which is considered normal, i.e., no contralateral deficits according to normative data [41].

## 5. Conclusions

This study, which includes a professional sample, contributes to increasing the knowledge of soccer players’ strength indicators, particularly in the lower body. Independently of age, a highly competitive level differentiates soccer players in the analyzed static strength, vertical jumping, and isokinetic strength metrics. This information is of great value to sports agents and coaches, underlining the need to design and include strength-specific content during soccer training to achieve higher competitive levels. Finally, it is recommended to develop future work that may consider longitudinal data regarding the variables analyzed.

## Figures and Tables

**Figure 1 ijerph-20-01799-f001:**
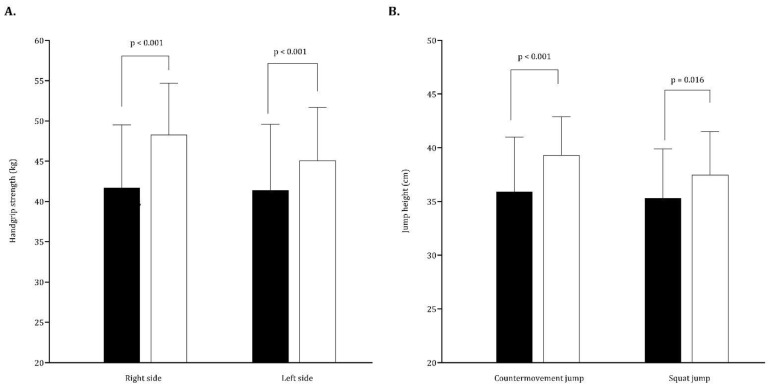
Means and standard deviations for handgrip strength (panel **A**) and jumping (panel **B**) by age group.

**Figure 2 ijerph-20-01799-f002:**
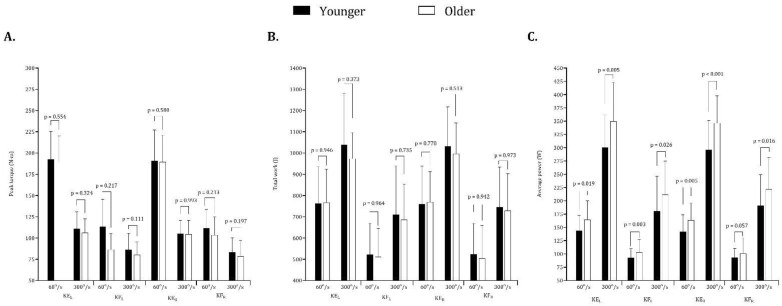
Means and standard deviations for peak torque (panel **A**), total work (panel **B**) and average power (panel **C**) by age group.

**Table 1 ijerph-20-01799-t001:** Summary of the isokinetic strength variables analyzed.

Variables	Units	Description
Peak torque	Newtons/meter (N·m)	Highest force output during a repetition. Indicative of muscle strength capabilities.
Total work	Joules (J)	The sum of work for every repetition performed. Represents a more functional value of muscle performance, as work is torque sustained over time in an isometric test.
Average power	Watts (W)	The average rate of doing work. This is how effectively the muscle can perform work over time.

**Table 2 ijerph-20-01799-t002:** Comparison of the static strength and vertical jumping according to players’ competitive level.

Variables	A Team (1)	B Team (2)	U-23 (3)	U-19 (4)	*p*	*
	Mean (SD)	Mean (SD)	Mean (SD)	Mean (SD)		
Strength indicators						
Handgrip right side (kg)	49 (4.6)	48.1 (7.6)	44.2 (8.4)	39.7 (6)	≤0.001	1 > 3 and 4; 2 > 4; 3 > 4
Handgrip left side (kg)	47.9 (6.5)	43.5 (6.6)	42.8 (8.1)	39.1 (6.8)	≤0.001	1 > 3 and 4; 2 > 4
CMJ height (cm)	40.7 (4.3)	38.2 (3.5)	37.2 (4.2)	34.5 (4.5)	≤0.001	1 > 2, 3 and 4; 2 > 4; 3 > 4
SJ height (cm)	38.4 (4.3)	37.3 (3.8)	35.3 (4.1)	34.7 (4.5)	0.019	1 > 3 and 4

SD, standard deviation; CMJ, countermovement jump; SJ, squat jump; * significant differences between groups.

**Table 3 ijerph-20-01799-t003:** Comparison of the isokinetic strength concentric movement performance according to players’ competitive level.

	Extension—Left Knee						Extension—Right Knee					
	A Team (1)	B Team (2)	U-23 (3)	U-19 (4)			A Team (1)	B Team (2)	U-23 (3)	U-19 (4)		
	Mean (SD)	Mean (SD)	Mean (SD)	Mean (SD)	*p*	*	Mean (SD)	Mean (SD)	Mean (SD)	Mean (SD)	*p*	*
Peak Torque												
60°/s (N·m)	198.0 (29.6)	177.7 (32.8)	177.2 (27.0)	206.5 (28.2)	0.007	1 > 3; 2 ≤ 4; 3 ≤ 4	196.3 (26.7)	186.3 (36.1)	171.1 (37.7)	205.1 (26.5)	0.056	3 ≤ 4
300°/s (N·m)	108.4 (14.2)	106.6 (20.3)	99.2 (15.1)	119.1 (19.3)	0.027	3 ≤ 4	106.4 (12.7)	105.0 (20.1)	96.6 (14.1)	111.0 (15.4)	0.082	n.s.
Total Work												
60°/s (J)	808.9 (156.6)	722.0 (157.2)	674.2 (142.0)	838.5 (151.8)	0.007	1 > 3; 2 ≤ 4; 3 ≤ 4	811.1 (142.5)	747.4 (151.9)	666.2 (144.5)	820.5 (163.0)	0.022	1 > 3; 3 ≤ 4
300°/s (J)	1010.9 (106.1)	954.0 (156.7)	882.2 (141.4)	1160.3 (229.8)	≤0.001	1 > 3; 2 ≤ 4; 3 ≤ 4	1019.6 (115.0)	1019.5 (185.0)	886.7 (126.9)	1124.8 (159.9)	0.001	1 > 3; 3 ≤ 4
Average Power												
60°/s (W)	172.4 (34.9)	155.8 (37.3)	154.1 (27.3)	134.1 (24.5)	0.011	1 > 4	167.5 (30.2)	163.7 (34.9)	151.6 (35.7)	130.3 (20.7)	0.002	1 > 4; 2 > 4
300°/s (W)	350.3 (37.7)	370.8 (109.0)	309.9 (45.6)	279.5 (58.9)	≤0.001	1 > 3 and 4; 2 > 4	353.1 (35.9)	355.0 (68.0)	313.9 (45.5)	270.0 (44.2)	≤0.001	1 > 3 and 4; 2 > 4; 3 ≤ 4
	**Flexion—Left Knee**						**Flexion—Right Knee**					
Peak Torque												
60°/s (N·m)	111.2 (17.8)	94.5 (23.0)	95.1 (15.4)	129.3 (34.8)	≤0.001	1 > 2 and 3; 1 ≤ 4; 2 ≤ 4; 3 ≤ 4	107.9 (18.2)	96.5 (25.0)	97.7 (16.0)	125.0 (18.4)	≤0.001	1 > 3 and 1 ≤ 4; 2 ≤ 4; 3 ≤ 4
300°/s (N·m)	82.1 (11.7)	81.5 (20.9)	77.9 (16.1)	91.2 (20.0)	0.179		78.9 (13.5)	81.3 (25.6)	75.8 (12.5)	87.2 (19.9)	0.419	
Total Work												
60°/s (J)	559.7 (129.5)	453.4 (141.9)	441.6 (104.6)	588.5 (132.5)	0.001	1 > 2 and 3; 2 ≤ 4; 3 ≤ 4	561.8 (141.7)	430.3 (167.2)	444.2 (90.3)	587.7 (140.8)	0.002	1 > 2 and 3; 2 ≤ 4; 3 ≤ 4
300°/s (J)	748.2 (164.0)	635.2 (176.6)	633.5 (124.3)	750.5 (277.3)	0.013	1 > 2 and 3; 3 ≤ 4	761.3 (139.7)	720.4 (224.5)	667.1 (128.4)	791.0 (216.5)	0.145	1 > 3
Average Power												
60°/s (W)	111.7 (20.8)	91.4 (27.3)	98.2 (16.2)	86.9 (15.6)	0.003	1 > 3 and 4	108.9 (23.6)	89.6 (35.7)	97.5 (19.2)	89.5 (14.3)	0.018	1 > 4
300°/s (W)	232.8 (60.7)	199.7 (72.6)	193.7 (40.8)	155.6 (65.5)	0.002	1 > 3 and 4	237.5 (51.9)	214.4 (70.8)	206.7 (47.6)	165.5 (54.1)	0.003	1 > 4; 2 > 4; 3 ≤ 4

SD, standard deviation; * significant differences between groups; n.s., non-significant.

**Table 4 ijerph-20-01799-t004:** Comparison of the static strength and vertical jumping performance according to players’ filed position.

Strength Indicators	GK (1)	DEF (2)	MID (3)	AT (4)	*p*	*
	Mean ± SD	Mean ± SD	Mean ± SD	Mean ± SD		
Handgrip right side (kg)	45.8 (9.5)	48.1 (7.6)	44.2 (8.4)	39.7 (6.8)	0.554	n.s.
Handgrip left side (kg)	43.9 (8.3)	43.5 (6.6)	42.8 (8.1)	39.1 (6.8)	0.398	n.s.
CMJ height (cm)	38.4 (5.2)	38.2 (3.5)	37.2 (4.2)	34.5 (4.5)	0.939	n.s.
SJ height (cm)	36.6 (3.8)	37.3 (3.8)	35.3 (4.1)	34.7 (4.5)	0.985	n.s.

SD, standard deviation; GK, goalkeeper; DEF, defender; MID, midfielder; AT, attacker; CMJ, countermovement jump; SJ, squat jump; * significant differences between groups; n.s., non-significant.

**Table 5 ijerph-20-01799-t005:** Comparison of isokinetic strength concentric movement performance according to players’ field position.

	Extension—Left Knee						Extension—Right Knee					
	GK (1)	DEF (2)	MID (3)	AT (4)			GK (1)	DEF (2)	MID (3)	AT (4)		
	Mean (SD)	Mean (SD)	Mean (SD)	Mean (SD)	*p*	*	Mean (SD)	Mean (SD)	Mean (SD)	Mean (SD)	*p*	*
Peak Torque												
60°/s (N·m)	202.9 (27.0)	194.4 (31.4)	182.8 (35.3)	194.5 (25.2)	0.287	n.s.	196.3 (26.7)	186.3 (36.1)	171.1 (37.7)	205.1 (26.5)	0.056	3 ≤ 4
300°/s (N·m)	117.2 (17.8)	107.4 (18.1)	104.4 (20.6)	113.3 (13.3)	0.190	n.s.	104.2 (8.4)	106.0 (15.6)	100.4 (17.1)	109.9 (15.6)	0.439	n.s.
Total Work												
60°/s (J)	857.9 (177.7)	757.6 (149.5)	744.3 (186.1)	782.3 (143.7)	0.352	n.s.	772.1 (146.1)	762.1 (161.0)	743.5 (156.9)	798.0 (174.3)	0.705	n.s.
300°/s (J)	1096.4 (189.4)	989.4 (173.4)	1002.2 (237.2)	1005.2 (130.4)	0.712	n.s.	984.7 (68.2)	1021.6 (171.8)	988.7 (180.8)	1042.9 (157.6)	0.854	n.s.
Average Power												
60°/s (W)	170.2 (34.4)	157.5 (35.2)	146.5 (32.1)	159.5 (34.1)	0.296	n.s.	160.2 (31.3)	155.2 (40.3)	145.3 (28.8)	161.0 (29.4)	0.355	n.s.
300°/s (W)	331.4 (37.7)	320.3 (69.5)	313.0 (52.9)	353.8 (98.8)	0.606	n.s.	303.1 (70.4)	327.1 (68.8)	316.4 (57.1)	332.1 (40.1)	0.744	n.s.
	**Flexion—Left Knee**						**Flexion—Right Knee**					
Peak Torque												
60°/s (N·m)	113.8 (15.6)	112.5 (35.7)	102.0 (21.3)	110.3 (22.4)	0.626	n.s.	113.3 (15.3)	106.8 (24.6)	101.8 (21.5)	114.8 (18.9)	0.173	n.s.
300°/s (N·m)	80.5 (8.7)	87.5 (17.4)	78.5 (19.0)	84.7 (16.1)	0.173	n.s.	74.6 (14.7)	86.1 (19.9)	76.7 (17.9)	80.4 (15.1)	0.234	n.s.
Total Work												
60°/s (J)	517.2 (135.4)	509.3 (145.4)	515.6 (140.6)	528.9 (143.7)	0.955	n.s.	513.5 (156.8)	499.7 (159.9)	516.9 (152.4)	528.0 (140.7)	0.964	n.s.
300°/s (J)	641.7 (247.9)	697.1 (205.1)	724.8 (204.3)	677.0 (172.8)	0.941	n.s.	589.7 (202.7)	767.5 (206.1)	734.8 (192.8)	739.6 (76.0)	0.377	n.s.
Average Power												
60°/s (W)	103.0 (29.8)	97.7 (22.6)	95.7 (20.6)	101.0 (22.4)	0.777	n.s.	104.8 (28.8)	95.3 (29.5)	95.6 (20.3)	101.1 (22.6)	0.696	n.s.
300°/s (W)	184.2 (87.4)	196.4 (77.8)	197.2 (45.5)	202.6 (70.4)	0.933	n.s.	169.2 (87.7)	213.7 (72.8)	206.3 (50.7)	210.7 (46.0)	0.755	n.s.

SD, standard deviation; GK, goalkeeper; DEF, defender; MID, midfielder; AT, attacker; * significant differences between groups; n.s., non-significant.

## Data Availability

The data presented in this study are available upon request from the corresponding author.

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
