# Peer review of "Characterization of Static Strength, Vertical Jumping, and Isokinetic Strength in Soccer Players According to Age, Competitive Level, and Field Position"

_ijerph, 2023, doi:10.3390/ijerph20031799_

Round 1
Reviewer 1 Report
Summary: this cross-sectional study aimed to compare different measures of muscle strength among soccer players according to age, competitive level, and filed position. To this aim, a large number of parameters were collected that describe 5 aspects of strength: static strength, peak torque, total work, average power, agonist/antagonist ratio. While the collection of data is impressive there are a number of severe problems that however could be dealt with eventually:
First, it is very difficult for the non-expert to discern the meaning of summary variables used in Tables 1-3, especially as torque, work, power, and the ratios are measured in the same places (’60 degree/sec left knee extension’ etc.). I strongly recommend an additional table describing in few words what is measured in each of the 5 cases, including units, and references. For instance, what distinguishes total work and average power, both usually measured in Watt? Are there several measurements on which average power is based? If possible, the last column in such a table could indicate which specific aspect of strength the respective measure stands for. Please choose the most appropriate description for each measure and use the same term throughout the manuscript.
Second, a large number of variables (n = 46) were compared between three sets of predicting variables, i.e. age, competitive level, and field position. A reduced significance value should be used, e.g. Bonferroni correction which gives p = 0.05/46 = 0.001, which would also help to see the main pattern of results. However, it would be preferable to combine the site-specific and/or site+type-specific (flexion/extension) into fewer summary variables that describe the main aspect of each variable group with less random error. This would certainly help to get a clearer picture and could eventually be presented in a forest plot or similar.
Third, the results show that competitive level shows the largest group contrasts, followed by age, while no differences are seen by field position. It is very likely that the age difference is explained by the higher age of team A players who also perform better regarding strength than players at lower competitive level. By definition, the players with least experience (U19), and lower strength, are all included in the younger category. The most interesting would have been to retrospectively collect strength parameters of A team players at younger ages, say age 20, and compare these with same-aged players in U19 or U23. If there was no difference, one could conclude that coaches should focus on strength development; if there are incumbent team A players already had superior strength, this might lead to other conclusions. But as such data are not available, the authors should perform multivariable modelling, e.g. set up a linear model for each outcome variable such as average handgrip, or average total work (see point 2) and use categories of competitive level and age (continuous or categorical) as predictors in the same model. There will be large differences by competitive level, and including age into the model will test, whether age can contribute an addition aspect or not.
Last, please rework introduction and discussion to make it more concise, and shorter.
Further comments:
Participants: where do the participants come from, country, town(s)? do the 4 groups include participants from different clubs? if yes, how is the affiliation with different clubs accounted for?
Methods: please describe in methods that age was dichotomized at 20.9 years and why.
Results: this is a cross-sectional study. When describing group differences please refrain from using expressions such as ‘increase in older players in power…’ (line 192), because the wording suggests that players are followed over time, and changes in performance within individuals are measured, which is not the case. Please always give the reference category when presenting results of a comparison.
Please give 1 or at most 2 non-zero digits for the p-value (e.g. p = 0.2 instead of 0.234, 0.014 instead of 0.0142).
Introduction, lines 80-87: ref.12 is cited with results that elite players performing better than amateurs, but the next sentence presents results on vertical jumping that suggest the opposite. Please clarify and give results for all groups that are compared, not only the amateur group. The last sentence of this paragraph is fine.
Methods: the notation of mean +/- SD suggests that the underlying variable is symmetrically distributed. This is however never shown. Please use mean (SD) instead, which is a valid description irrespective of distribution.
Tables 1-3: Please add units to all variables.
Line 192: problem in the sentence
Author Response
Manuscript ID: ijerph-2078366
Reviewer 1.
Summary: this cross-sectional study aimed to compare different measures of muscle strength among soccer players according to age, competitive level, and field position. To this aim, a large number of parameters were collected that describe 5 aspects of strength: static strength, peak torque, total work, average power, and agonist/antagonist ratio. While the collection of data is impressive there are a number of severe problems that however could be dealt with eventually:
- First, it is very difficult for the non-expert to discern the meaning of summary variables used in Tables 1-3, especially as torque, work, power, and the ratios are measured in the same places (’60 degree/sec left knee extension’ etc.). I strongly recommend an additional table describing in a few words what is measured in each of the 5 cases, including units, and references. For instance, what distinguishes total work and average power, both usually measured in Watt? Are there several measurements on which average power is based? If possible, the last column in such a table could indicate which specific aspect of strength the respective measure stands for. Please choose the most appropriate description for each measure and use the same term throughout the manuscript.
Response 1: We thank the reviewer’s overall positive feedback on our work, and we have followed the advice regarding the summary of the variables used. Therefore, we added one Table in the Methods section describing the variables considered for analysis (Please see Table 1).
- Second, a large number of variables (n = 46) were compared between three sets of predicting variables, i.e. age, competitive level, and field position. A reduced significance value should be used, e.g. Bonferroni correction which gives p = 0.05/46 = 0.001, which would also help to see the main pattern of results. However, it would be preferable to combine the site-specific and/or site+type-specific (flexion/extension) into fewer summary variables that describe the main aspect of each variable group with less random error. This would certainly help to get a clearer picture and could eventually be presented in a forest plot or similar.
Response 2: The reviewer is correct. We restructure all the results considering the main important isokinetic strength metrics (i.e., Peak Torque, Total Work, and Average Power). We also focus our analysis on 60º and 300º/s to decrease the number of variables in the tables. All the tables were reorganized based on site+type-specific (flexion/extension). Finally, we also calculated the Bonferroni corrections in Table 1 and Table 2.
- Third, the results show that competitive level shows the largest group contrasts, followed by age, while no differences are seen by field position. It is very likely that the age difference is explained by the higher age of team A players who also perform better regarding strength than players at lower competitive level. By definition, the players with least experience (U19), and lower strength, are all included in the younger category. The most interesting would have been to retrospectively collect strength parameters of A team players at younger ages, say age 20, and compare these with same-aged players in U19 or U23. If there was no difference, one could conclude that coaches should focus on strength development; if there are incumbent team A players already had superior strength, this might lead to other conclusions. But as such data are not available, the authors should perform multivariable modelling, e.g. set up a linear model for each outcome variable such as average handgrip, or average total work (see point 2) and use categories of competitive level and age (continuous or categorical) as predictors in the same model. There will be large differences by competitive level, and including age into the model will test, whether age can contribute an addition aspect or not.
Response 3: As suggested by the reviewer, a standard multiple regression was conducted to calculate the variance in each strength indicator that can be explained by age, competitive level, and player's field position. In the model, the competitive level was the strongest predictor of handgrip right side (beta= -0,63; p<0,001), handgrip left side (beta= -0,60; p<0,001), and CMJ height max (beta= -0,36; p=0,029). The variance in each of these variables explained by the model ranged between 21% and 25%. Between isokinetic strength metrics, in the average power, again the competitive level made the strongest variation on all metrics analyzed (-0,43 < Betas > -0,76; ps<0,004) after controlling for age. The variance in each average power angle velocity explained by the model ranged between 22% and 33%. This full paragraph was added to the end of results section.
- Last, please rework the introduction and discussion to make it more concise, and shorter.
Response 4: Several editing was made across the introduction and discussion section as suggested.
- Participants: where do the participants come from, country, town(s)? do the 4 groups include participants from different clubs? if yes, how is the affiliation with different clubs accounted for?
Response 5: The four groups (teams) presented in this study were from the same club. Overall, players were from 14 different countries (Portugal = 56, Brazil = 10, Iran = 1, Mozambique = 2, Colombia = 4, Angola = 1, Cameroon = 1, Nigeria = 2, Sweden = 1, Italy = 1, Tunisia = 1, Ghana = 1, England = 1, Brunei = 1). We have followed to reviewer’s advice and added more detailed information concerning the participants as follows: “The A team was competing at the First Portuguese League, the B team was competing at the Fourth Portuguese National Division, and the U23 and U19 teams were competing in the main National Division for their age groups. Overall, players were from 14 different countries, however, most of them (n = 56) were Portuguese. All teams were part of the same club. The assessments were conducted at the beginning of the sports season and all participants were healthy at the time of data collection.”
- Methods: please describe in methods that age was dichotomized at 20.9 years and why.
Response 6: Age was dichotomized into two categories according to the Median (20.9). In our study, we decided to build two Age categories according to the Median (20.9) in order to have a bigger number of players being compared. Also, to better explore the contribution of age on the variation of all strength metrics analyzed, in the revised manuscript we performed regression analysis using age as a continuous variable.
- Results: this is a cross-sectional study. When describing group differences please refrain from using expressions such as ‘increase in older players in power…’ (line 192), because the wording suggests that players are followed over time, and changes in performance within individuals are measured, which is not the case. Please always give the reference category when presenting the results of a comparison.
Response 7: We agree with the reviewer's comment, and we have updated the sentence as follows: “Older players showed significantly better results than younger players in the average power on left knee extension (60º/s and 300º/s), right knee extension (60º/s and 300º/s), left knee flexion (60º/s and 300º/s), and at 300º/s right knee flexion.”
- Please give 1 or at most 2 non-zero digits for the p-value (e.g. p = 0.2 instead of 0.234, 0.014 instead of 0.0142).
Response 8: We have upgraded all the p-values in the same way.
- Introduction, lines 80-87: ref.12 is cited with results that elite players performing better than amateurs, but the next sentence presents results on vertical jumping that suggest the opposite. Please clarify and give results for all groups that are compared, not only the amateur group. The last sentence of this paragraph is fine.
Response 9: We agree with the reviewer's comment, and we have updated the sentence as follows: “In soccer, a previous study comparing knee flexor muscle strength considering the players’ competitive level, reported that elite players presented stronger hamstrings than their non-elite and amateur peers [12]. Additionally, the literature has described the superior jumping ability of elite (49,9 ± 7,5 cm) compared to non-elite soccer players (43,9 ± 6,9 cm) [13]. These results underline that individuals involved in the highest competitive levels tend to outperform their lower divisions counterparts in lower-body strength assessments.”
- Methods: the notation of mean +/- SD suggests that the underlying variable is symmetrically distributed. This is however never shown. Please use mean (SD) instead, which is a valid description irrespective of distribution.
Response 10: As suggested by the reviewer, we have updated the notation of mean ± SD to mean (SD) throughout the manuscript.
- Tables 1-3: Please add units to all variables.
Response 11: We added units to all variables in the Tables.
- Line 192: problem in the sentence
Response 12: The sentence has been updated following the reviewer’s previous suggestions.
Reviewer 2 Report
It would be necessary to identify the competitive level of each group. It would also be necessary to specify at what time of the season the data were taken.
On the other hand, for comparisons by age, the number of subjects in each group should be specified. Likewise, it would be necessary to justify why it is precisely ages older or younger than 20.9 years that are compared.
Limb results should be expressed as a function of laterality dominance. In this sense, right and left values cannot be interpreted in the same way for a right-handed person as for a left-handed person.
If the objective is the characterization of a population, the results should be clearer. There are too many data that do not provide information on the object of study and are purely descriptive. The extensive tables should be reconsidered and some graphs should be used to help characterization.
The analysis by positions does not identify the number of subjects in each group. This number will probably be small and will not contribute to the reliability of the comparison between groups. In this sense, table 3 shows the same variables as table 2 and there are significant differences in only one variable. Wouldn't it be better to eliminate that table? or perhaps the comparison between positions should be questioned?
This type of obvious phrases should be eliminated:
These results may reflect the different strategies used in the specific field position training sessions and the type of physical demands required for each field function/position.
Overall, the results emphasize the superior strength levels of professional soccer players compared with their lower-division peers.
Author Response
Manuscript ID: ijerph-2078366
Reviewer 2.
- It would be necessary to identify the competitive level of each group. It would also be necessary to specify at what time of the season the data were taken.
Response 1: We have followed the reviewer’s advice and added more information regarding participants as follows: “The A team was competing at the First Portuguese League, the B team was competing at the Fourth Portuguese National Division, and the U23 and U19 teams were competing in the main National Division for their age groups. Overall, players were from 14 different countries, however, most of them (n = 56) were Portuguese. All teams were part of the same club. The assessments were conducted at the beginning of the sports season and all participants were healthy at the time of data collection.”
- On the other hand, for comparisons by age, the number of subjects in each group should be specified. Likewise, it would be necessary to justify why it is precisely ages older or younger than 20.9 years that are compared.
Response 2: In our study, we decided to build two Age categories according to the Median (20,9) to have a bigger number of players being compared. Also, to better explore the contribution of age on the variance of all strength metrics analyzed, in the revised manuscript we performed regression analysis using age as a continuous variable.
- Limb results should be expressed as a function of laterality dominance. In this sense, right and left values cannot be interpreted in the same way for a right-handed person as for a left-handed person.
Response 3: Unfortunately, we don’t have this information about lateral dominance. As we don’t have this information available, we decide to plot all the information in table 3 considering age, we calculated the bilateral differences. Our conclusion, considering the peak torque measurement, is that the bilateral differences ranged between 0,2% to 5,2%, which is considered normal, i.e., no contralateral deficits (Sapega, 1990).
- If the objective is the characterization of a population, the results should be clearer. There are too many data that do not provide information on the object of study and are purely descriptive. The extensive tables should be reconsidered, and some graphs should be used to help characterization.
Response 4: We agree with the reviewer's suggestion. Also, Reviewer 1 suggested better organizing the information on the tables and reducing some information. Keeping this in mind. We restructure all the results considering the main important isokinetic strength metrics (i.e., Peak Torque, Total Work, and Average Power). We also focus our analysis on 60º and 300º s to decrease the number of variables in the tables. All the tables were reorganized based on site+type-specific (flexion/extension).
- The analysis by positions does not identify the number of subjects in each group. This number will probably be small and will not contribute to the reliability of the comparison between groups. In this sense, table 3 shows the same variables as table 2 and there are significant differences in only one variable. Wouldn't it be better to eliminate that table? or perhaps the comparison between positions should be questioned?
Response 5: We partially agree with the reviewers about the small number of participants in each group. In this study, the smaller group was the goalkeeper (n=6). The rest of the groups are not so small: Defender (n= 27); midfielder (n=30); attacker (n=20). To deal with this limitation on the sample size in each group, we performed non-parametric statistics. We believe this information by field function/position is very important to optimize some training procedures considering the specificity of some field positions/functions. This could help coaches better individualize training programs since the load required during the game is not the same, considering the different field positions/functions.
- This type of obvious phrases should be eliminated:
These results may reflect the different strategies used in the specific field position training sessions and the type of physical demands required for each field function/position.
Overall, the results emphasize the superior strength levels of professional soccer players compared with their lower-division peers.
Response 6: As suggested, we eliminated the previously mentioned sentences from the text.
Reviewer 3 Report
Dear Authors
You have written an interesting study. However, some parts need to be improved for greater clarity.
Abstract: - report statistical analysis and p values in/of significant results
Introduction:
The part relating to isokinetic and CMJ the following paper could be used in the introduction and to strengthen the discussion: 10.3233/IES-182138
Otherwise, the rationale to choose the particular test is clearly presented.
Methods
Report the playing experience of participants and other possible exclusion criteria like musculoskeletal injuries.
Please report generally at what part of the day was the testing done, what was the order of tests and what was the break between tests.
Line 131 - only 30s between jumps? Back up this decision with references.
For CMJ and SJ procedures add references
What was the rationale to test all 3 speeds with isokinetic?
Line 166- 5 practice trials - at each speed or in the CPM mode? report
Which leg was tested first? How long was the break for other leg? Report
Also, report which variables did you take to further analysis.
Report effect size in your results.
Table 1 - units are missing by isokinetic values
Please report peak torque to body weight as absolute values can be misguiding.
Overall the discussion is short and could expand on the vastly football available research in this field/ just a few examples that could be helpfull - 10.3233/ies-2012-00483; DOI: 10.18276/cej.2022.3-01; DOI: 10.3233/IES-200240;
The conclusion heading is missing and the limitations paragraph should be the last paragraph of the discussion.
Overall an interesting study that needs some more work from the authors. Therefore, I recommend a major revision.
Kind regards
Author Response
Manuscript ID: ijerph-2078366
Reviewer 3.
- Dear Authors, you have written an interesting study. However, some parts need to be improved for greater clarity. Abstract: - report statistical analysis and p values in/of significant results
Response 1: We thank the reviewer’s overall positive feedback on our work. Several editing was made to the abstract related to statistical analysis and p values.
- Introduction: The part relating to isokinetic and CMJ the following paper could be used in the introduction and to strengthen the discussion: 10.3233/IES-182138. Otherwise, the rationale to choose the particular test is clearly presented.
Response 2: We appreciate the reviewer's suggestion. We integrate this reference in the introduction as well as in the discussion section to underline the relationship between isokinetic strength and jump performance.
- Methods: Report the playing experience of participants and other possible exclusion criteria like musculoskeletal injuries.
Response 3: Unfortunately, we do not have access to the individual playing experience, and we will integrate this information in future data collection moments. All the assessments were performed at the beginning of the season by players who were healthy, and no player was excluded due to injury. This information was integrated into the text, as follows: ”The assessments were conducted at the beginning of the sports season and all participants were healthy at the time of data collection.”
- Please report generally at what part of the day was the testing done, what was the order of tests and what was the break between tests.
Response 4: As suggested, we added this information in the text: “Participants were tested for static strength, vertical jumping, and muscular strength. During four consecutive days, each team was evaluated in a physical performance laboratory following the mentioned testing order. Between tests, a 5-minute break was given to each participant to avoid the effects of accumulated fatigue.”
- Line 131 - only 30s between jumps? Back up this decision with references. For CMJ and SJ procedures add references
Response 5: Although there is a more recent indication that confirms 1-min passive rest between jumps to ensure muscular recovery (Petrigna et al., 2019), the issue is not consensual in the literature. In our study, due to time restrictions to perform all the assessments in the protocol, especially in the professional team, we considered a period of 30s between the performance of the jump. Some studies also have presented fewer recovery times such as 20 s (Wu et al., 2019) and 30s between each repetition of CMJ (Ishak et al., 2022).
Petrigna, L., Karsten, B., Marcolin, G., Paoli, A., D'Antona, G., Palma, A., & Bianco, A. (2019). A Review of Countermovement and Squat Jump Testing Methods in the Context of Public Health Examination in Adolescence: Reliability and Feasibility of Current Testing Procedures. Frontiers in physiology, 10, 1384. https://doi.org/10.3389/fphys.2019.01384
Wu PP-Y, Sterkenburg N, Everett K, Chapman DW, White N, Mengersen K (2019) Predicting fatigue using countermovement jump force-time signatures: PCA can distinguish neuromuscular versus metabolic fatigue. PLoS ONE 14(7): e0219295. https://doi.org/10.1371/ journal.pone.0219295 -à 20 s CMJ
Ishak A, Wong FY, Seurot A, Cocking S, Pullinger SA (2022) The influence of recovery period following a pre-load stimulus on physical performance measures in handball players. PLoS ONE 17(3): e0249969. https://doi.org/10.1371/ journal.pone.0249969
- What was the rationale to test all 3 speeds with isokinetic?
Response 6: According to the literature, the 60 and 180º/s testing procedures are commonly used to reveal the highest force output, indicating a muscle strength capability, to which we agree. On the other hand, maximal power could be attained at higher angular velocities. Our idea in this study was to evaluate the maximum strength production capacity. The slow angular speeds are sufficient and recommended; thus, we analyzed 60º/s. Also, to analyze the force-velocity curve, since most actions in soccer are determined by maximal power, we analyzed 300º/s. The following references justify our options.
Menzel, H. J., Chagas, M. H., Szmuchrowski, L. A., Araujo, S. R., de Andrade, A. G., & de Jesus-Moraleida, F. R. (2013). Analysis of lower limb asymmetries by isokinetic and vertical jump tests in soccer players. The Journal of Strength & Conditioning Research, 27(5), 1370-1377.
Rouis, M., Coudrat, L., Jaafar, H., Filliard, J. R., Vandewalle, H., Barthelemy, Y., & Driss, T. (2015). Assessment of isokinetic knee strength in elite young female basketball players: correlation with vertical jump. J Sports Med Phys Fitness, 55(12), 1502-8.
- Line 166- 5 practice trials - at each speed or in the CPM mode? report
Response 7: The practice trials were performed at each speed. This information was added in the text: “Five trials were performed at each speed before each isokinetic test to ensure the correct execution”.
- Which leg was tested first? How long was the break for other leg? Report
Response 8: The preferred leg was tested first, followed by the non-preferred leg after a 3-minute break. This information was included in the text “The preferred leg was tested first, followed by the non-preferred leg under the same conditions and after a 3-minute break.”
- Also, report which variables did you take to further analysis.
Response 9: We appreciate the reviewer’s feedback, and we added this information as follows: “The values of Peak Torque, Total Work, and Average Power, were used for analysis. A more detailed description of these variables is presented in Table 1 according to the manufacturers’ manual [26].”
- Report effect size in your results.
Response 10: As suggested, we added the effect size when required.
- Table 1 - units are missing by isokinetic values
Response 11: We have included the units for isokinetic variables in the Tables.
- Please report peak torque to body weight as absolute values can be misguiding.
Response 12: In this study, we are focused on the assessment of the maximum force production capacity. To do so, we analyze the maximum moment of force, the total work related to the muscle's ability to produce force throughout the full range of motion, and average power, which is the dynamic parameter that better describes quick strength. Due to the vast number of metrics in our study's analyses, we decided not to include the peak torque to body weight as the absolute value. We acknowledge this limitation of the study.
- Overall the discussion is short and could expand on the vastly football available research in this field/ just a few examples that could be helpfull - 10.3233/ies-2012-00483; DOI: 10.18276/cej.2022.3-01; DOI: 10.3233/IES-200240;
Response 13: as suggested, some of the suggested references were added to the text.
- The conclusion heading is missing, and the limitations paragraph should be the last paragraph of the discussion.
Response 14:The recommendation was followed in the text.
- Overall an interesting study that needs some more work from the authors.
Response 15: We appreciate the reviewer’s positive feedback and overall feedback. We have updated the manuscript following the reviewers’ suggestions and we believe that its quality has improved.
Round 2
Reviewer 2 Report
I still think that some graphic would have improved the characterization that was sought.
Thank you for your review
Reviewer 3 Report
Dear Authors,
Thank you for addressing all of my comments. The paper's quality and clarity improved. Therefore, I recommend acceptance in its current form.
Kind regards